# Monitoring Time-Non-Stable Surfaces Using Mobile NIR DLP Spectroscopy

**Marek Gąsiorowski [1], Piotr Szymak [2,\*], Aleksy Patryn [1] and Krzysztof Naus [2]**

[1] Faculty of Electronics and Computer Science, Koszalin University of Technology, 75-453 Koszalin, Poland; gasiorowski.marek@o2.pl (M.G.); aleksy.patryn@tu.koszalin.pl (A.P.)

[2] Polish Naval Academy, Federation of Military Academies, 81-127 Gdynia, Poland; k.naus@amw.gdynia.pl

\* Correspondence: p.szymak@amw.gdynia.pl

**Abstract:** In recent years, Near Infrared (NIR) spectroscopy has increased in popularity and usage for different purposes, including the detection of particular substances, evaluation of food quality, etc. Usually, mobile handheld NIR spectroscopy devices are used on the surfaces of different materials, very often organic ones. The features of these materials change as they age, leading to changes in their spectra. The ageing process often occurs only slowly, i.e., corresponding reflection spectra can be analyzed each hour or at an even longer interval. This paper undertakes the problem of analyzing surfaces of non-stable, rapidly changing materials such as waxes or adhesive materials. To obtain their characteristic spectra, NIR spectroscopy using a Digital Light Projection (DLP) spectrometer was used. Based on earlier experiences and the current state of the art, Artificial Neural Networks (ANNs) were used to process spectral sequences to proceed with an enormous value of spectra gathered during measurements.

**Keywords:** NIR DLP spectroscopy; reflectance time-non-stable spectra; artificial neural network

## 1. Introduction

Nowadays, Near Infrared (NIR) spectroscopy has become very popular. Modern computers with more and more computing power and appropriate data analysis software make it possible to obtain a powerful tool that enables efficient processing of thousands of data points to obtain useful information about a tested sample. NIR spectroscopy has many significant advantages: the non-destructive nature of the test, no need to prepare a sample, simplicity and speed of measurements, and low cost. Some disadvantages bring limitations: the useful information is not directly available, and the sample may be described by thousands of variables. It often turns out that the spectra obtained are not repeatable [1].

Today, there are several examples of using NIR spectroscopy being used for various purposes, including the evaluation of food quality. One of them is to use a portable NIR spectrometer working in the range of 1396–2396 [nm] to collect the spectra of breast milk samples for quality evaluation [2], which is an essential matter for newborn children. The authors use different chemometrics to calculate and then develop 18 calibration models with and without using derivatives and the standard normal variate. Once the calibration models were developed, the best treatments were selected according to the correlation coefficients and prediction errors. The other example of using NIR to estimate food quality is included in [3]. The authors examined Visible (VS) and NIR spectroscopy usage to monitor grape composition within a vineyard to facilitate the decision-making process with regards to grape quality sorting and harvest scheduling. Measurements of grape clusters were acquired in the field using a VS/NIR spectrometer, operating in the 570–990 [nm] spectral range, from a motorised platform moving at 5 km/h. To analyse the obtained spectra, they used classical methods, e.g., a correlation function. In [4], the authors examined a novel prototype NIR instrument designed to measure dry matter content in single potatoes. The instrument is based on interaction measurements to measure deeper into the potatoes. It

measures rapidly, up to 50 sizes per second. The device also enables several distances to be recorded for each measurement. The instrument was calibrated based on three different potato varieties, and the calibration measurements were done in a process plant, making the calibration model suitable for production line use.

The other type of NIR spectroscopy usage is connected with the need to detect undesired substances. One of the examples is non-destructive detection of tomato pesticide residues using VS/NIR spectroscopy and prediction models such as ANNs [5]. The authors used VS/NIR spectral data from 180 samples of non-pesticide tomatoes (used as a control treatment) and samples impregnated with a pesticide with a concentration of 2/1000 [L], recorded by a spectroradiometer working in the range 350–1100 [nm], to train and then verify ANNs. The other example is included in [6]. The authors used NIR spectroscopy and characteristic variables selection methods to develop a quick way of determining cellulose, hemicellulose, and lignin contents in Sargassum horneri, i.e., the species of brown macroalgae that is common along the coast of Japan and Korea. Calibration models for cellulose, hemicellulose, and lignin in Sargassum horneri were established using partial least square regression methods with full variables. The last example showed the detection of measuring vitamin C and ellagic acid in wild-harvested Kakadu plum fruit samples [7]. The results of this study demonstrated the ability to predict vitamin C and ellagic acid in whole and pureed Kakadu plum fruit samples using a handheld NIR spectrophotometer. In the next paper [8], the NIR spectroscopy method was developed to analyze the oil and moisture contents of the plant Camellia gauchowensis Chang and C. semiserrata Chi seeds kernels. The authors used principal component analysis (PCA) and partial least squares (PLS) regression methods for calibration and validation. Finally, they obtained correlation coefficients of 0.98 and 0.95 for oil, and 0.92 and 0.89 for moisture, respectively, for calibration and validation.

More often, NIR spectroscopy is used with other methods to obtain the expected solution. One example is presented in [9]. The authors used NIR spectroscopy coupled with chemometric tools and obtained a fast and low-cost alternative solution for evaluating wood properties and quality categories. The obtained results of research showed that NIR spectroscopic data combined with powerful multivariate statistic tools and artificial intelligence solutions provided a fast and reliable tool, helpful in the decision-making process. The other opportunities are connected with NIR-absorbing organic semiconductors, especially for organic photovoltaic cells (OPV) [10]. OPV has increased its popularity in the field of renewable energy due to its lightweight, flexibility, and relatively low cost. To find new OPV materials, experiments with different types of NIR materials as active layers were conducted. Good results have also been achieved using NIR spectroscopy in conjunction with machine learning methods [11]. In this study, the authors used VIS/NIR spectroscopy for the effective discrimination of genetically modified (GM) and non-GM Brassica napus, B. rapa, and F1 hybrids (B. rapa X GM B. napus). As a classification method, the convolutional ANNs were used with success. More references to the advances in NIR spectroscopy and related computational methods can be found in [12,13].

Progress in the development of spectroscopy has led to the creation of a class of miniaturized spectrometers, including the VS and NIR [14,15]. The miniaturization of devices and systems is related to the tendency to perform non-laboratory measurements, including the adaptation of sizes to the in-line version [16,17]. An essential advantage of using smaller instruments is the potential possibility of implementing distributed measurement schemes and approaches to "remote" monitoring of environmental and "field" measurements.

As can be seen over a dozen different examples, the use of NIR spectroscopy, often supported by other classical methods and artificial intelligence methods, has been analyzed. This proves the great potential of NIR spectroscopy supported by other methods for data analysis. The factor connecting the above examples of NIR spectroscopy applications are the relatively slow processes of change taking place in the tested materials and substances, usually demanding monitoring every hour or at even longer intervals. Therefore, in this article the problem of using NIR spectroscopy supported by ANNs for the analysis of

rapidly changing processes of the transition from liquid to solid is discussed. The authors of this article did not find any other article dealing with this subject using the commonly used NIR spectroscopy. Waxes and adhesives commonly used in everyday life were selected as test objects. As shown in [18], features of the materials change in different environment conditions, e.g., ambient temperature. In this paper, the method of analyzing optical reflection spectra of objects whose properties change quickly over time using the Digital Light Projection (DLP) measurement technique was proposed. The surfaces of several different materials were selected as test objects, the optical properties of which can vary even within a few seconds and which can be first approximated as prototypes of materials used in electron technology. At the current stage, these test objects were: quick-drying glue, two-component epoxy glue, and natural beeswax. Spectroscopic measurements were carried out using the DLP NIRSCAN Nano EVM Spectrometer by Texas Instruments. To process the obtained spectra, ANNs were used. The earlier results of the research with the ANNs are included in [19,20].

The test measurements were carried out to estimate the possibilities and effectiveness of further mobile spectroscopic measurements of the surface of materials, emphasising mobility and the possibility of carrying out measurements in various conditions, especially apart from stationary laboratories presented.

A method of measuring reflectance with the spectrometer window positioned practically in direct contact with the tested surface was chosen. In-line measurement results were saved in .dat or .xlms files for processing with Matlab. The recording time of one reflectance spectrum in the 900–1700 nm range was 2.67 s, while the spectrum was recorded six times and averaged.

The kit version of DLP NIRSCAN NANO and the primary measurement schemes were used in the same way as included in [21,22]. The modernisation of the device allowed it to work more efficiently after using its own housing design. The test object in the configuration is typically placed on the windows at the top of the monochromator.

In the next section, the applied measurement method and stand were described. Then, the results of measurements were presented. At the end, the discussion on received results and final conclusions were included.

## 2. Methods

The spectra were recorded using DLP NIRSCAN NANO, i.e., a small-size Texas Instrument spectrometer operating in the light wavelength range from 900 to 1700 nm with optical resolution equal to 10 nm [23]. The measurements were carried out several times at a given measuring point, depending on the selected option. The dimensions of the device, equal to $58 \times 62 \times 36$ mm allowed us to perform mobile studies of reflectance spectra. The device can work in the reflective mode. In Figure 1, the general view and data flow in the measurement stand are visualised. The hardware part of the stand consists of the DLP NIR EVM spectrometer connected with a PC using USB. The spectrometer produces light of a specific wavelength illuminating the examined sample and then measures the intensity of light that passes through the sample. The software part of the stand includes the DLP NIRscan Nano GUI (used to control the device, simple visualisation of measurements, collecting and saving of data in the appropriate format) and the Matlab program (used for data processing employing ANNs). The spectra obtained from the first software can be displayed and saved in two formats: CSV and DAT. Matlab accepts both file formats.

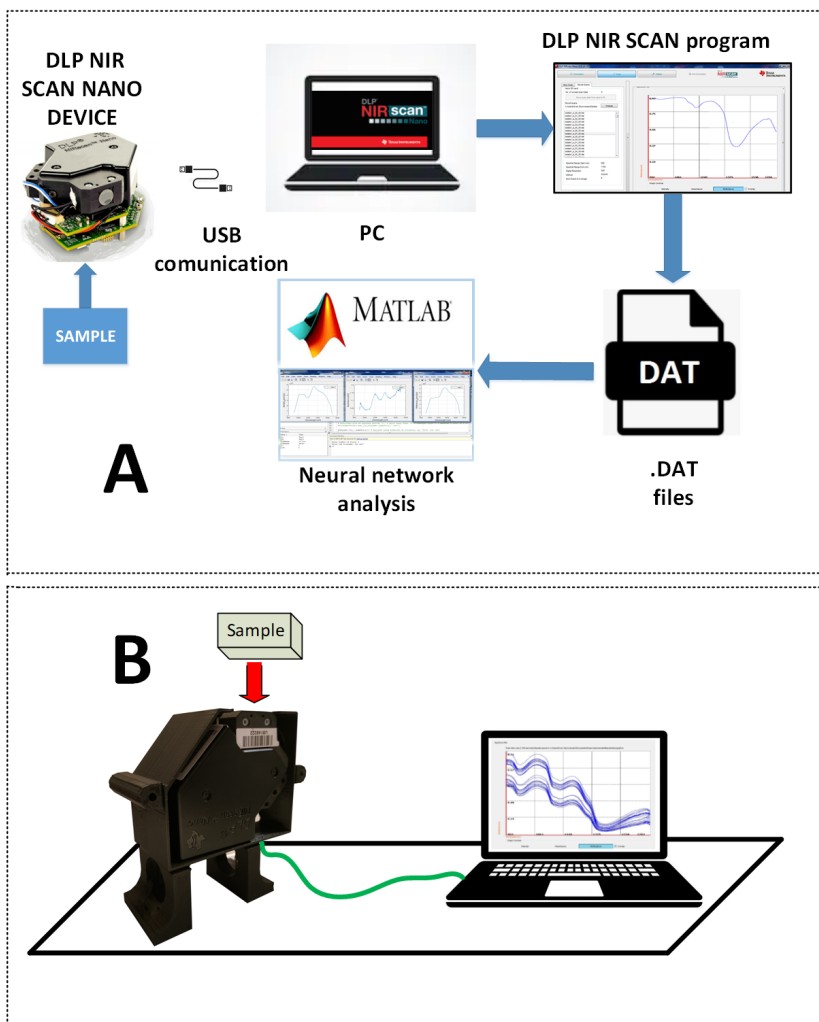

**Figure 1.** The measurement stand based on the DLP NIR EVM spectrometer: (**A**)—a scheme of the data flow, (**B**)—a general view of the stand.

## 3. Results

The research results were divided into two groups, depending on the type of examined materials:

1. Wax materials,
2. Adhesive materials.

Both types of materials changing from liquid to solid (solidification time) at different rates. Therefore, the research was conducted with different time scales to observe the essential parts of the processes.

The measurements results of both material groups are presented in the two following subsections. In the next subsection, the results of research using Artificial Neural Networks (ANNs) are included.

### 3.1. Wax Materials

Paraffin and beeswax were selected as representatives of the first group of materials. These materials are characterised by a quick transition from liquid to solid phase of approx. 30 [s]. Therefore, the time interval between consecutive measurements equal to 2.67 [s] was chosen. In Figure 2, the obtained results of seizing reflectance *R* for paraffin and natural beeswax for subsequent time steps and different wavelengths of light are illustrated. The reflectance is given without a physical unit. If *R* achieves 1, it means that all the light was reflected. At first glance, the transition of the paraffin is similar to beeswax. However, as it

turned out, the spectra of these two substances differed significantly in reflectance levels during the change of the state of aggregation. Still, the characteristics are very similar. Both materials have characteristic peaks around 1200 [nm] and 1400 [nm]. Some differences between the materials may be because due to their different colours, i.e., the tested beeswax is yellow while the paraffin is white. The total observation time ranged from 0 to 28.6 s, during which time the materials changed from liquid to solid. The value $t_1$ is equal to 0 [s], and $t_{11}$ is equal to 28.6 [s].

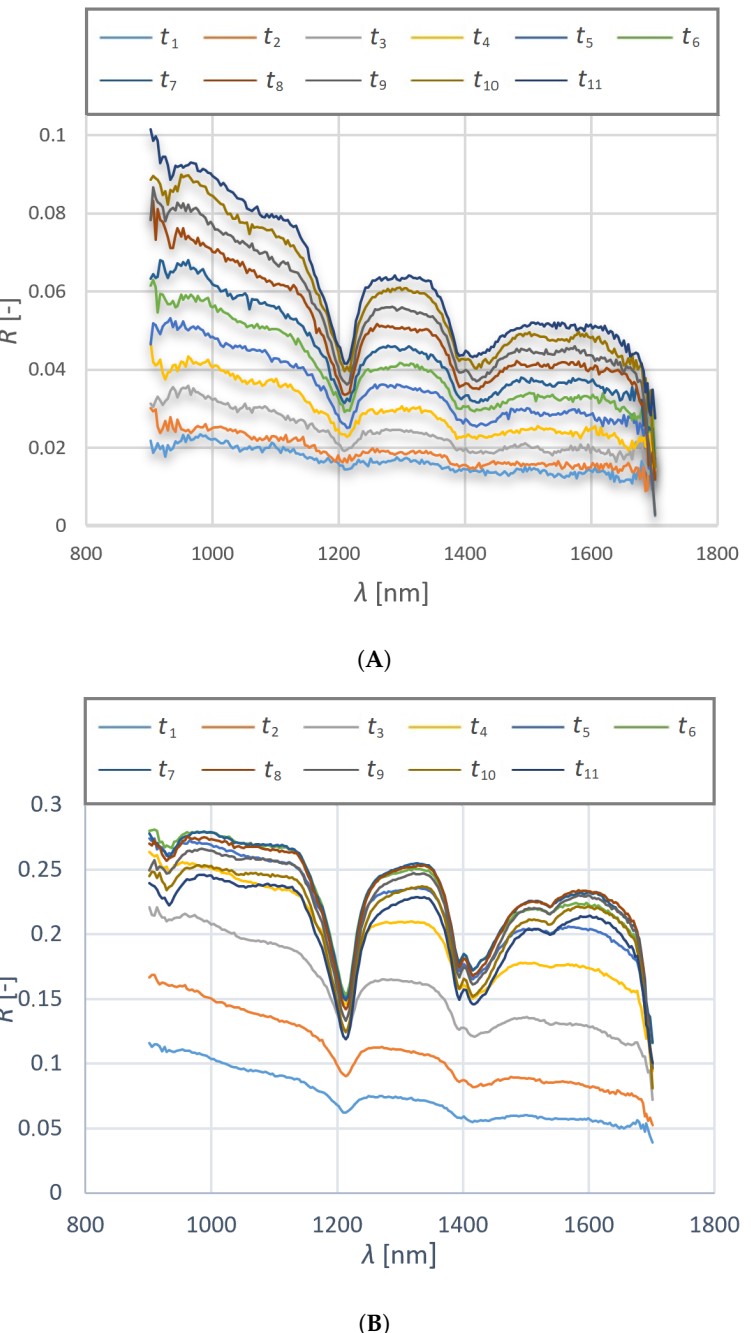

**Figure 2.** Reflectance spectra of two wax materials: (**A**)—paraffin, (**B**)—beeswax.

### 3.2. Adhesive Materials

The next group of tested materials were adhesives. General-purpose adhesives available on the commercial market and used in households. The two following kinds of such adhesives were selected and then investigated using DLP NIR spectrometry:

1.  Cyanoacrylate glue, so called "superglue",
2.  Two-component epoxy glue.

Due to the longer time of transition from liquid to solid phase of the second group of materials, a longer time interval between successive measurements was selected, i.e., 60 [s]. All the measurements took 10 [min]. In Figure 3, the results of changes in reflection spectra over time were illustrated, i.e., over the course of the solidification process. The changes during the superglue test turned out to be not very expressive. Even after hardening, the glue was characterised by high transparency, and no apparent changes over time could be observed. Significant differences were observed only at the edge of the measurement ranges. A considerable fluctuation can be seen in the long-wave region. It may be due to the operating range of the device. A more extensive measuring range would allow a more precise material analysis.

Considering the reflectance spectrum of epoxy glue (Figure 3), similar to wax materials, characteristic peaks around 1200 [nm] and 1400 [nm] can be observed. After a detailed comparison, it should be stated that the characteristics peaks are more distant from each other than for the wax materials, i.e., the first peak can be seen below a wavelength of 1200 [nm] and the second at wavelengths higher than 1400 [nm].

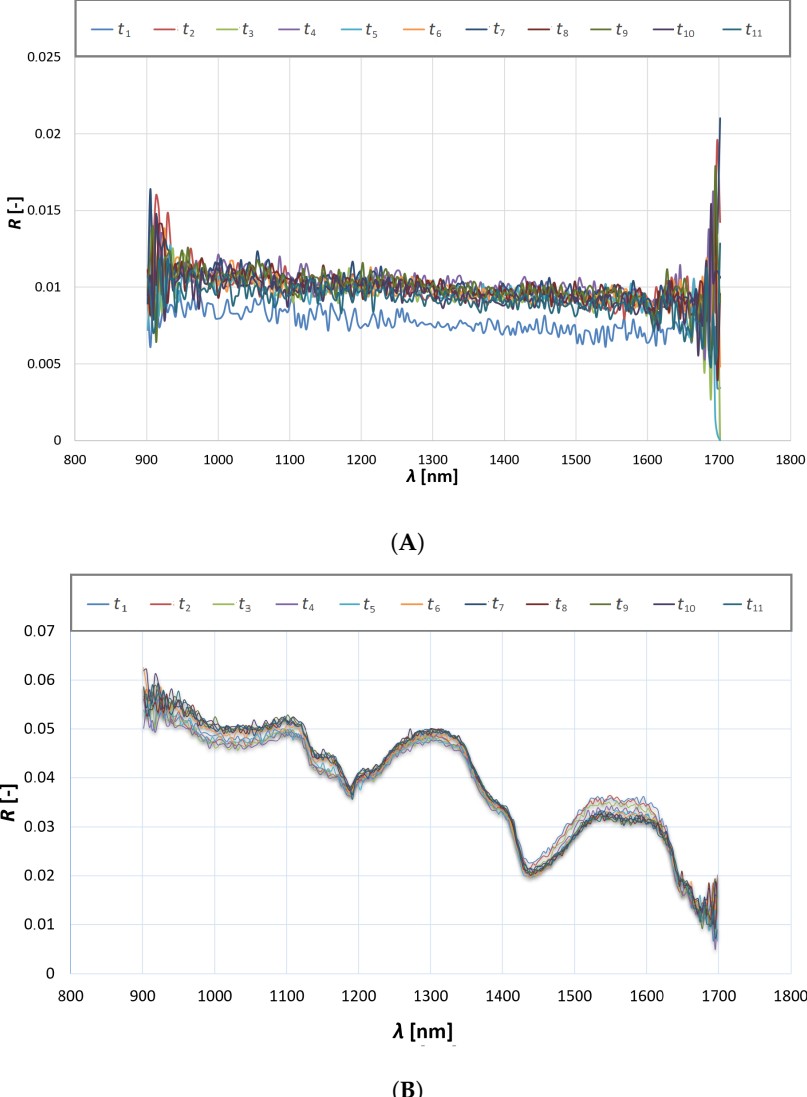

**Figure 3.** Reflectance spectra of two adhesive materials: (**A**)—superglue, (**B**)—epoxy glue.

Moreover, it is tough to differentiate the lines corresponding to the successive time steps. The lines are overlapped, and their order changes for various wavelength ranges, e.g., the range from 900 [nm] to 1430 [nm] and the range from 1430 [nm] to 1700 [nm]. It is possible that analysis of the adhesive materials requires a spectrometer offering a broader wavelength.

The obtained in the following time steps reflectance spectra of selected waxes and adhesives showed that the spectra course and their change over time are unique for each material. It can be used for determining relationships between specific spectra and the progress of the solidification process of the particular material. However, the measurement of adhesives needs to use a spectrometer offering a wider wavelength than used in research. It results from the observations that it is tough to differentiate the lines corresponding to the successive time steps. Moreover, the lines are overlapped and their order changes for various wavelength ranges.

### 3.3. Different Variants of ANNs

The obtained results (reflection spectra of individual materials) were uploaded to the Matlab program (Figure 4) to train the neural networks based on the obtained series of measurements, which can be used to identify the solidification stage (stage) of the material observed. However, in the first place, it was necessary to check the efficiency of the available learning methods and the correct minimum number of hidden neurons $n_h$ to minimise the Mean Square Error (MSE). This is essential because the level of this error tells us about the degree of network training (the ability to recognise objects with a relatively low time error).

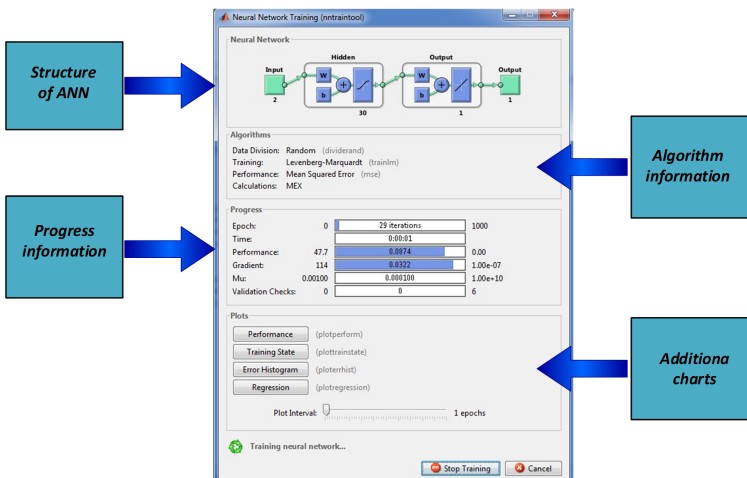

**Figure 4.** Matlab toolbox for ANN training and verification.

Different training methods for ANNs available in Matlab were also compared to find the method which reduces the MSE error to a minimum with the lowest possible number of hidden neurons. For example, the analysis for a network containing 200 neurons using a typical PC (4 GB operating memory, 3.30 GHz processor) takes about 2 h. During the tests, eleven methods of learning the network were tested based on up to 80 measurement spectra for each of the selected test objects.

The general structure of ANNs is illustrated in Figure 5. ANNs consists of three inputs, a hidden layer including the specific number of neurons, and one output layer. As can be seen, the number of neurons in the hidden layer was changed from 10 to 200 at step 10 of the research.

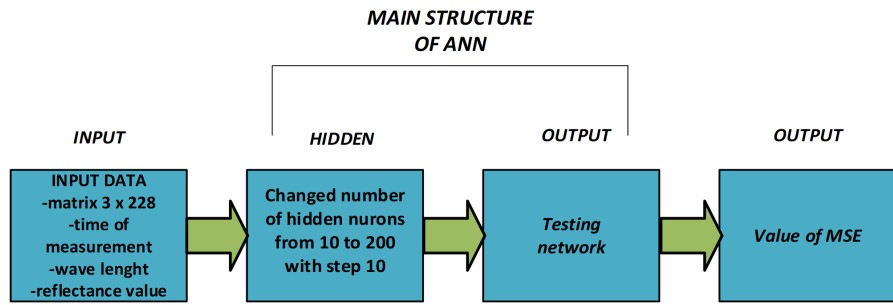

**Figure 5.** General structure of ANNs.

## 4. Discussion

In Figure 6, the ANNs verification results have been presented. The ANNs were trained by different learning methods on two measurement series for beeswax and paraffin materials. For the appropriate methods, the network in the first case achieves a satisfactory level of MSE with about 40 hidden neurons, which directly translates into a short analysis time. However, when analysing paraffin, the course of the MSE change is quite different, despite their being similar materials. A satisfactorily low error rate is obtained with about 20 hidden neurons. The mathematical description of the learning methods, using abbreviated names from Matlab, is included in [24]. In Table 1, the abbreviated names are presented with the corresponding full name of the methods.

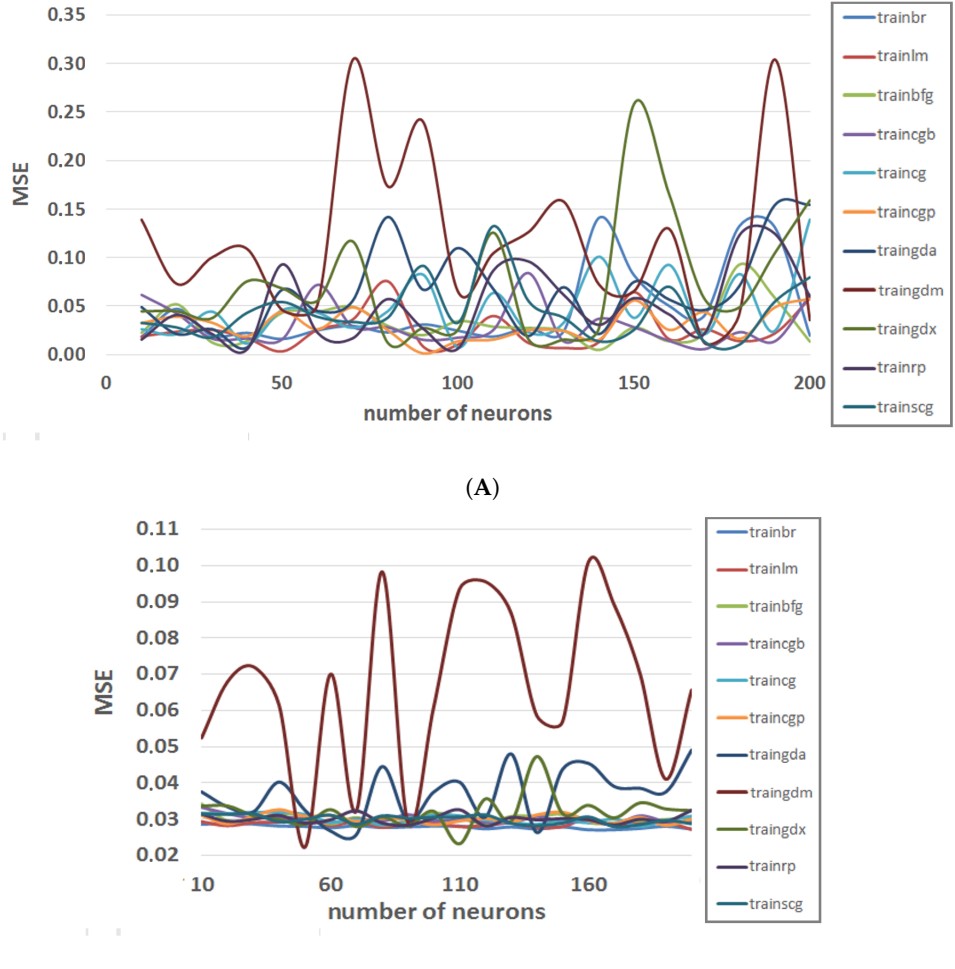

**Figure 6.** MSE for ANN's verification for different learning methods and number on neurons in the hidden layer for wax materials: (**A**)—beeswax, (**B**)—paraffin.

**Table 1.** The abbreviated and full names of the learning methods for ANNs.

| No of Method | Abbreviated Name | Full Name |
|---|---|---|
| 1 | trainbr | Bayesian regularization backpropagation |
| 2 | trainlm | Levenberg–Marquardt backpropagation |
| 3 | trainbfg | BFGS quasi-Newton backpropagation |
| 4 | traincgb | Conjugate gradient backpropagation with Powell–Beale restarts |
| 5 | traincgf | Conjugate gradient backpropagation with Fletcher–Reeves updates |
| 6 | traincgp | Conjugate gradient backpropagation with Polak–Ribiére updates |
| 7 | traingda | Gradient descent with adaptive learning rate backpropagation |
| 8 | traingdm | Gradient descent with momentum backpropagation |
| 9 | traingdx | Gradient descent with momentum and adaptive learning rate backpropagation |
| 10 | trainrp | Resilient backpropagation |
| 11 | trainscg | Scaled conjugate gradient backpropagation |

The trangdm method (Gradient descent with momentum backpropagation) achieved a lower error rate with about 50 hidden neurons (Table 2). Nevertheless, the course of MSE changes as a function of the number of neurons is chaotic. The network learning method that achieves a significantly lower MSE rate for the value of around 110 hidden neurons is traingdx (gradient descent with momentum and adaptive learning rate backpropagation) method. An essential piece of information resulting from comparing these two allegedly similar materials is that, as can be observed, increasing the number of hidden neurons does not significantly improve the MSE value. It only causes a significant extension of the analysis time. In addition, it turns out that the bottom learning method optimal for a given material (measurement series of reflection spectra) will not be suitable for another measurement series of a different material, even for materials supposedly similar to each other.

**Table 2.** The obtained minimum MSE and corresponding number of neurons $n_h$ in the hidden layer for the selected learning methods and the wax materials.

| No of Method | Beeswax: MSE | Beeswax: $n_h$ | Paraffin: MSE | Paraffin: $n_h$ |
|---|---|---|---|---|
| 1 | 0.016 | 50 | 0.027 | 120 |
| 2 | 0.003 | 50 | 0.027 | 200 |
| 3 | 0.005 | 140 | 0.028 | 80 |
| 4 | 0.006 | 170 | 0.029 | 60 |
| 5 | 0.007 | 100 | 0.028 | 180 |
| 6 | 0.001 | 90 | 0.028 | 100 |
| 7 | 0.007 | 40 | 0.026 | 70 |
| 8 | 0.012 | 170 | 0.022 | 50 |
| 9 | 0.012 | 80 | 0.023 | 110 |
| 10 | 0.006 | 40 | 0.028 | 90 |
| 11 | 0.010 | 180 | 0.028 | 140 |

The idea for the further development of the research is presented in Figure 7. The use of the statistical Principal Component Analysis (PCA) method is also considered, which would allow for a reduction in the statistical data set and speed up the analysis time. The idea behind the algorithm is to monitor and save the most significant variances in the data set. Elements with low or zero variance of the set are ignored [25]. However, as already mentioned, for a given measurement series of a specific material, it will be necessary to

select a particular method and determine the optimal number of hidden neurons to shorten the analysis time as much as possible.

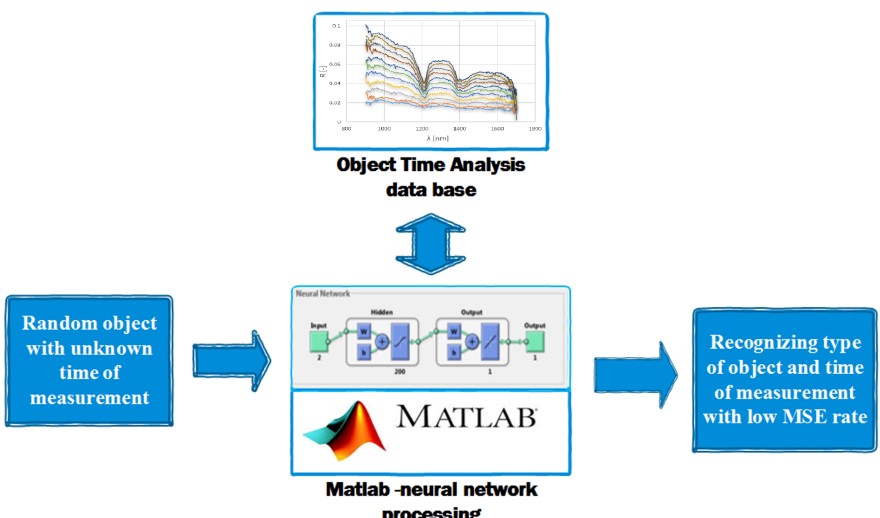

**Figure 7.** Conception of future development of software in Matlab.

## 5. Conclusions

Measurements of optical properties of materials whose aggregate state is unstable can be effectively carried out using the DLP method using DPL NIR Scan equipment. To increase the certainty of analysis of rapidly changing spectra, the method of spectra analysis using neural networks was tested. Four materials were tested in the study: epoxy, cyanoacrylate glue, paraffin, and beeswax. The first two materials are adhesives, while the second two are waxes. Based on the results of the research, the following conclusions can be stated:

1. Samples from the adhesive materials showed almost no changes in the optical spectrum in the range from 900 to 1700 nm during the solidification process. Therefore, their further analysis using neural networks is pointless. The lack of visible changes will not allow determining of the characteristic features for a given moment of the solidification process. A spectrometer with a wider wavelength range would possibly give better results.
2. The results of measurement of waxes showed significant changes in optical properties in the measurement range mentioned above. Along with the solidification process, the reflectance level changes slightly and the characteristic shape changes.
3. ANN analysis showed MSE values for individual learning methods with a relatively small number of 40–50 neurons at satisfactory levels. It is possible to describe sets of spectra using ANNs with appropriate learning methods, reducing the number of hidden neurons to a minimum. This is vital because it significantly affects the analysis time.

Thanks to the DLP NIRSCAN Nano device, it is possible to monitor the reflection spectra and thus the changes taking place in various types of materials. Analysing the spectra makes it possible to observe the course and dynamics of changes, which may be helpful information during technological processes where the given materials are used. The analysis of the spectra of objects with variable optical parameters using ANNs tells us that the described learning methods implemented in Matlab can learn entire series of measurements without significant problems and then recognise specific measures while maintaining a relatively small error (the described learning methods used about 80% of the measurement series for learning and about 20% of the series were used for network verification and MSE determination). During the analysis, it turned out that there is no universal method of training the networks, even for similar materials. The method and the

optimal number of neurons should be matched to a given material, which in the future may allow for the recognition of materials or determining their condition. The next planned stage of the study is the modification of the program. After reading the given measuring series of the spectra of a given object, it will be possible to determine the time and stage of setting of a random measured adhesive sample. Thanks to this solution, it will be possible to determine the time (solidification stage) with a small error in any technological processes where wax is used. Such information can be beneficial in solidifying glues, resins, and other materials.

**Author Contributions:** Conceptualization, M.G. and A.P.; methodology, M.G. and A.P.; software, P.S. and K.N.; validation, M.G. and P.S.; formal analysis, A.P.; investigation, M.G.; resources, M.G.; data curation, M.G.; writing—original draft preparation, M.G. and P.S.; writing—review and editing, P.S.; visualization, M.G. and P.S.; supervision, A.P.; project administration, A.P.; funding acquisition, P.S. and K.N. All authors have read and agreed to the published version of the manuscript.

**Funding:** This research received no external funding.

**Institutional Review Board Statement:** Not applicable.

**Informed Consent Statement:** Not applicable.

**Data Availability Statement:** Not applicable.

**Acknowledgments:** Authors thank Leszek Bychto for useful discussion and consultations.

**Conflicts of Interest:** The authors declare no conflict of interest.

## Abbreviations

The following abbreviations are used in this manuscript:

| | |
|---|---|
| ANN | Artificial Neural Network |
| CSV | Comma-Separated Values |
| DAT | generic DATa file |
| DLP | Digital Light Projection |
| DMD | Digital Micromirror Device |
| EVM | Evaluation Module |
| MSE | Mean Square Error |
| NIR | Near Infrared Spectroscopy |
| PC | Personal Computer |
| USB | Universal Serial Bus |
| VS | Visible |

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
