# Peer review of "Monitoring Time-Non-Stable Surfaces Using Mobile NIR DLP Spectroscopy"

_electronics, doi:10.3390/electronics11131945_

Round 1

Reviewer 1 Report

Abstract: The aim of this study was to....The aim of this study 'is' to

spectrometr---typo error

within a few minutes or several dozen minutes....this is vague

Abstract is not well written..It doesn't give a clear picture of the proposed work. Must we re-written and must be little more detailed.

Section 2: The relevance of figure 1 is not clear. What novelty does it carry?

"The resolution of the measuring range was 228x228 pixels, i.e. every 3.5 nm."...sentence is not clear

There is not literature survey and the article jumps to results directly from methods...no problem statement has been defined.

The paper has not been written with serious and novel contributions. Require a thorough revision and restructuring.

Author Response

Dear Reviewer,

Subject: revision of the paper entitled "Monitoring time-non-stable surfaces using mobile NIR DLP spectroscopy"

Thank you for your comments and suggestions. We appreciate the time and details provided by you and have incorporated the suggested changes into the manuscript to the best of our ability. The manuscript has certainly benefited from these insightful revision suggestions. Our responses are given in a point-by-point manner below.

We hope that the manuscript after extensive revision is now suitable for publication and look forward to hearing from you in due course.

Sincerely,

Piotr Szymak

Prof. Assoc.

Polish Naval Academy

  • Abstract: The aim of this study was to....The aim of this study 'is' to

The abstract was rewritten, including the above remark.

  • spectrometr---typo error

According to the remark, it was corrected.

  • within a few minutes or several dozen minutes....this is vague

According to the suggestion, the vague wording was changed to "a few seconds".

  • Abstract is not well written..It doesn't give a clear picture of the proposed work. Must we re-written and must be little more detailed.

The Abstract was rewritten to give a clearer picture of the proposed research with a more detailed description.

  • Section 2: The relevance of figure 1 is not clear. What novelty does it carry?

Figure 1 was deleted.

  • "The resolution of the measuring range was 228x228 pixels, i.e. every 3.5 nm."...sentence is not clear

The sentence was deleted, and the previous sentence was completed with information about optical resolution received from the DLP® NIRscan™ Nano EVM User's Guide. The corrected sentence has the following form: "The spectra were recorded using DLP NIRSCAN NANO, i.e. a small-size Texas Instrument spectrometer operating in the light wavelength range from 900 to 1700 nm with the optical resolution equal to 10 nm [23]."

We apologise for the misleading information.

  • There is not literature survey and the article jumps to results directly from methods...no problem statement has been defined.

The problem statement was precisely formulated and inserted in the Introduction section before the presentation of the results.

  • The paper has not been written with serious and novel contributions. Require a thorough revision and restructuring.

Additional explanations about novelty were included in both the Abstract and Introduction sections. The main novelty is connected with developing and verifying the method of analysing optical reflection spectra of materials whose properties change very quickly over time using the Digital Light Projection (DLP) measurement technique and Artificial Neural Networks (ANNs).

Reviewer 2 Report

 Journal:

Manuscript ID:

Title: Monitoring a time-non-stable surfaces using mobile NIR DLP spectroscopy

Authors: Marek Gasiorowski , Piotr Szymak , , Aleksy Patryn  and Krzysztof Naus

I want to thank the journal and also the editor who send to me this paper. I read the paper very carefully. I have summarized the report in the following steps

Reviewer comments

The aim of this study was to develop a method of analyzing non-stable surfaces based on a mobile Near Infrared (NIR) spectroscopy using Digital Light Projection (DLP) spectrometr and analysis of optical reflection spectra. The surfaces of several different materials were selected as prototype objects. Their optical properties can change even within a few minutes or several dozen minutes. The Artificial Neural Networks (ANNs) were used to process spectral sequences to proceed with an enormous value of spectra gathered during measurements.,

I recommend authors answer the following comments:

1.      Why literature need new mythologies in the presence of many other significant approaches?. And add these two recent studies

https://www.sciencedirect.com/science/article/abs/pii/S0577907317314922

https://www.sciencedirect.com/science/article/pii/S2211379720300723

2.      Why developed method in this paper is more reliable than others.

3.      Why authors are doing this work? What is the hurdle encountered by this study?

4.      The authors used two materials Wax materials and Adhesive materials in the present study, why?

5.      Authors claim that, ‘’Both kinds of materials differ, especially their clotting time’’ what is clotting time of both materials?

6.      In figure 3, the impact of time is discussed for Wax materials, what is physical significance?

7.      In figure 3, the impact of time is discussed for Adhesive materials, what is physical significance?

8.      Can authors predict, which one is more useful and why?

9.      Can authors present the comparison between Wax materials and Adhesive materials in context of time? 

10.  What are the major findings? Add into conclusion.

11.  Discussion and conclusion should be in different sections for better understanding.

After these corrections I recommend this paper for publication.

Author Response

Dear Reviewer,

Subject: revision of the paper entitled “Monitoring time-non-stable surfaces using mobile NIR DLP spectroscopy”

Thank you for your comments and suggestions. We appreciate the time and details provided by you and have incorporated the suggested changes into the manuscript to the best of our ability. The manuscript has certainly benefited from these insightful revision suggestions. Our responses are given in a point-by-point manner below.

We hope that the manuscript, after extensive revision is now suitable for publication and look forward to hearing from you in due course.

Sincerely,

Piotr Szymak

Prof. Assoc.

Polish Naval Academy

  1. Why literature need new mythologies in the presence of many other significant approaches?. And add these two recent studies

https://www.sciencedirect.com/science/article/abs/pii/S0577907317314922

https://www.sciencedirect.com/science/article/pii/S2211379720300723

The main novelty of the approach described in the paper is connected with developing and verifying the method of analysing optical reflection spectra of objects whose properties change very quickly over time. This method uses the Digital Light Projection (DLP) measurement technique and Artificial Neural Networks (ANNs). Additional explanations about novelty were included in both the Abstract and Introduction sections.

The first study was added to the state-of-the-art analysis.

  1. Why developed method in this paper is more reliable than others.

     The paper is not focused on a comparison of different methods. It is devoted to developing and verifying the method of analysing optical reflection spectra of objects whose properties change very quickly over time. The method was verified earlier for the materials with a relatively stable ageing process. The great advantage of this method may be that the measuring station only contains a handheld computer and a spectroscope, i.e. the station is mobile.

  1. Why authors are doing this work? What is the hurdle encountered by this study?

The idea of doing research included in the paper is related to the willingness to verify the method NIR spectroscopy supported by ANNs to evaluate materials with parameters changing rapidly over time. The main hurdle during this study was connected with the need of quick operation by the spectrometer. The additional burden was related to the large number of data received from the measurements.

  1. The authors used two materials Wax materials and Adhesive materials in the present study, why?

     We searched for materials with a short solidification time to test our method. Paraffin and beeswax were representatives of waxes, which characterise a short solidification time approx. 30 s. While the epoxy glue and cyanoacrylate glue are representatives of adhesives, which transfer from liquid to solid more quickly in approx. 5 minutes. This selection allows us to verify our method for materials with fast-changing features.

     Moreover, thanks to analysing the solidification process of given materials, using ANN in the future will enable us to reverse the analysis process to recognise the stage of the solidification process of a given material.

  1. Authors claim that, ‘’Both kinds of materials differ, especially their clotting time’’ what is clotting time of both materials?

Incorrect usage of the term “clotting time” is a language mistake. This sentence was changed to “Both kinds of materials differ in the rate of transferring from liquid to solid (solidification time)”.

  1. In figure 3, the impact of time is discussed for Wax materials, what is physical significance?

The NIR spectroscopy enables us to determine the reflectance spectra of the examined material’s surface. The spectra change over time during the solidification of the measured materials due to the changing reflectance properties. The relationship between the spectrum and the duration of the solidification process can be reused, e.g. in industry.

  1. In figure 3, the impact of time is discussed for Adhesive materials, what is physical significance?

     The physical significance of adhesives is similar to the importance indicated for waxes. It can be seen that different spectra were received for each wax and each adhesive.

  1. Can authors predict, which one is more useful and why?

We are not sure if we correctly understood the question. If the question is connected with a comparison between waxes and adhesives, it is worth underlining that the results received for waxes are more valuable. It is related that it is tough to differentiate the lines corresponding to the successive time steps for adhesives. The lines are overlapped, and their order is changed for various wavelength ranges, e.g. the range from 900 [nm] to 1430 [nm] and the range from 1430 [nm] to 1700 [nm]. Possible, the analysis of the adhesives materials needs to use a spectrometer offering a wider wavelength. Therefore, the data obtained from waxes measurement were used in the further study, i.e. using ANNs.

  1. Can authors present the comparison between Wax materials and Adhesive materials in context of time?

According to the suggestion, the additional paragraph including a comparison between waxes and adhesives in the context of the time was inserted at the end of section 3.2.

10.What are the major findings? Add into conclusion.

     The main findings were placed in a detailed list in the Conclusion section to highlight them.

11.Discussion and conclusion should be in different sections for better understanding.

     According to the suggestion, discussion and conclusions were inserted in different sections.

Round 2

Reviewer 1 Report

The authors have revised the manuscript considerably and addressed the review comments satisfactorily.